# Knowledge mobilisation: an ethnographic study of the influence of practitioner mindlines on atopic eczema self-management in primary care in the UK

Fiona Cowdell

This paper presents independent research funded by the National Institute for Health Research.

Faculty of Health Education and Life Sciences, Birmingham City University, Birmingham, UK

**Correspondence to**
Professor Fiona Cowdell;
fiona.cowdell@bcu.ac.uk

## ABSTRACT

**Objective** To explore how atopic eczema specific mindlines are developed by primary care practitioners.

**Design** Ethnographic study.

**Setting** One large, urban general practice in central England.

**Participants** In observation, all practitioners and support staff in the practice and in interviews a diverse group of practitioners (n=16).

**Results** Observation of over 250 hours and interview data were combined and analysed using an ethnographic approach through the lenses of mindlines and self-management. Three themes were identified: beliefs about eczema, eczema knowledge and approaches to self-management. Eczema mindlines are set against a backdrop of it being a low priority and not managed as a long-term condition. Practitioners believed that eczema is simple to manage with little change in treatments available and prescribing limited by local formularies. Practice is largely based on tacit knowledge and experience. Self-management is expected but not often explicitly facilitated. Clinical decisions are made from knowledge accumulated over time. Societal and technological developments have altered the way in which practitioner mindlines are developed; in eczema, for most, they are relatively static.

**Conclusions** The outstanding challenge is to find novel, profession and context-specific, simple, pragmatic strategies to revise or modify practitioner mindlines by adding reliable and useful knowledge and by erasing outdated or inaccurate information thus potentially improve quality of eczema care.

## INTRODUCTION

Atopic eczema (hereafter 'eczema') is a common, long-term skin condition affecting around one in five children and one in 12 adults in the UK. It can have a detrimental impact on well-being and quality of life and globally is one of the 50 most burdensome diseases.[1] Eczema is mainly treated in primary care.[2] People may seek advice from general practitioners (GPs), practice nurses (PNs), nurse practitioners, health visitors (HVs),

**Strengths and limitations of this study**

► First ethnographic study to examine the development of atopic eczema specific mindlines.
► Diverse sample primary care practitioners.
► Ethnographer was a lone researcher.
► Results may be context specific.

community pharmacists (CP) and pharmacy counter staff (PCS).

GP consultations are often unsatisfactory for both patient[3] and practitioner[4 5] with GPs dominating encounters and using avoidance tactics[6] and there being significant dissonance between patient/parent and GP beliefs about assessment and treatment.[6] Many GPs have limited specialist dermatology knowledge.[7] Nurse consultations, albeit in secondary care, tend to be more positively evaluated[8 9] and minimal research has been conducted into the contribution of HVs. Research into the role of the CP in dermatology care is limited[10] and expertise may be suboptimal[11] despite CPs reports of being at least reasonably confident in their role.[12] The role of the pharmacy counter assistant is equally under-researched although they are often first point of contact for customers and may offer health advice independent of pharmacists.[10 13]

The mainstay of eczema treatment is the regular application of emollients, at least daily and often for many years, with or without intermittent topical steroids and calcineurin inhibitors. Non-adherence results from the high self-management demand of applying topical treatments[14] but also lack of information and conflicting advice from different health professionals.[15] Despite available evidence (for example the Global Resource for Eczema Trials database[16] and the National Institute for Health and Care Excellence,

Guideline for Eczema[17]) providing evidence-based treatment appears to be a challenge for health professionals managing eczema.[18]

Self-management is a policy imperative which can improve disease outcomes and quality of life for people living with long-term conditions.[19] Strategies to support eczema self-management are poorly understood, have limited availability, can be costly and have variable impact.[20] Eczema is not classified as a long-term condition in the same way as other illnesses for example asthma.[21]

Primary care practitioners are expected to deliver evidence-based practice (EBP). Evidence-based medicine was originally the preserve of doctors and was defined as 'the conscientious, explicit and judicious use of current best evidence in making decisions about the care of individual patients'.[22] Over time, other professions have embraced EBP but this has, at times, been conceptualised as a set of research-based facts which if disseminated to practitioners will ensure more standardised, high quality care[23]; this notion is now largely dismissed.[24] Primary care practitioners face particular challenges in EBP given the volume of information they need and information overload is a real problem.[25]

The study of knowledge mobilisation (KM) is growing exponentially in healthcare, at its simplest it is 'moving knowledge to where it can be most useful'.[26] KM involves determined efforts to create, share and use research and other forms of knowledge predicated on the understanding that to be effective KM activity must be relational, constructed from social interaction and context-specific.[27–29]

Mindlines, developed from a primary care based ethnographic study[30] offer a 'real world' approach to mobilising knowledge and changing clinical practice. Mindlines are 'collectively reinforced, internalised tacit guidelines' which underpin clinical decision-making.[30] They build on the work of Polanyi[31] and Nonaka and Takeuchi[32] who propose that knowledge is not necessarily conscious and explicit, and that tacit knowledge in the form of unconscious schemata and technical know-how, are dominant influencers of action compared with formal codified knowledge. Gabbay and le May[28] suggest that mindlines are based on flexible, embodied and intersubjective understanding of knowledge that is grounded in the acceptance that there are multiple realities and that knowledge is context-specific. Mindlines represent a complex amalgamation of knowledge gathered from many sources for example, communication with colleagues and opinion leaders in the field and from tacit knowledge developed over time.[28] In their original work, Gabbay and le May[30] examined the construction of mindlines across primary care. A subsequent synthesis of 10 years of mindline literature (n=340) reports that they have been conceptualised and used in four distinct ways. 'Nominal' in which the term was used in name only, sometimes with a degree of scepticism, 'in practice' examining how mindlines are developed and spread in everyday practice, 'theoretical and philosophical' in which the

aim was to extend existing theory and 'solution focused', exploring ways in which mindlines can be influenced. Solution focused papers (n=28) emphasise the importance of collaborative learning, relationship building and effective leadership in the development of valid, collective, evidence-based mindlines. This review reveals a paucity of information about development or strategies to amend condition specific mindlines.[33] Repeating the search strategy utilised for this review in 2018 revealed an abundance of further related literature but little directly addressing condition specific mindlines or how they may best be amended.

Given the prevalence of eczema, the challenges of primary care consultations and the high self-management demand, it is prudent to investigate the way in which eczema mindlines are constructed by practitioners. This will inform understanding of mindlines 'in practice' and will underpin future 'solution focused' work to develop novel, context-specific, simple and pragmatic strategies to revise or modify eczema mindlines by adding reliable and useful knowledge and by erasing outdated or inaccurate information, thus potentially improve quality of eczema care and self-management.

## METHOD

### Aim
To understand construction of healthcare practitioner atopic eczema mindlines in primary care.

### Design
An ethnographic approach was employed. Ethnography is founded in anthropology and is concerned with the systematic study of people and cultures.[34] Data is collected through extensive observation with informal conversations, field notes and interviews.[35 36] Data was collected in one large general practice in England.

### Setting, participants and process
Data were collected by the author, a nurse and researcher, from January 2017 to June 2017. The general practice was identified by a local clinical research network. It was a research and education active urban general practice in a demographically diverse and deprived area of England with a patient population of approximately 10 000. Observations were also conducted in a community pharmacy adjacent to the practice, which was used by most patients. No practitioners reported a special interest in dermatology. In preparation for data collection the researcher attended two practice meetings to outline conduct of the study. Data were collected in more than 250 hours of observation during all surgery opening hours. The role of social-participant-as-observer, that is, predominantly observer with some social functions such as cleaning couches was taken.[37] Observation began with the reception team to understand the day-to-day working of the practice. Observation of consultations with GPs, GP trainees and locums, nurses,

## Box 1   Practitioner interview topic guide

► Do you have any special interest in skin health?
► How much contact do you have with patients with eczema?
► What sort of treatments do you use most often?
► How do you decide on a particular treatment?
► What impact does the local formulary have on your prescribing?
► How much are you able to advise patients on how to care for their eczema?
  – Concordance, etc.
► How do you update your own knowledge about eczema?
► How could we best get research information to use in your practice?
  – What methods do you use now?
  – Can you give any specific examples?
► Do patients come with their own ideas about the treatment they need?
► How much do you and your patient share the decision about what treatment to use?
► How do you reconcile patient's needs with what is available?
► Do you refer patients to any external sources of information?

## Box 2   Complete data set

**Observations and informal interviews**
► One general practice.
► Ten sessions observing reception and waiting room.
► Nine sessions observing in baby clinics.
► Two sessions observing in community pharmacy.
► Twenty-four sessions observing general practitioners (GPs).
► Five sessions with practice manager.
► Multiple informal meetings and one-to-one informal discussions.
► Four practice meetings.
► Six debriefs with GP trainees.
► Sixteen formal interviews, details provided in table 1.

**Documentary sources**
► Local prescribing guidelines.
► Online guidance accessed by practitioners during observation.

health visitors in baby clinics, held on the practice premises and pharmacy staff followed. GP telephone consultations were listened to and discussed with the practitioner. Field notes were documented and informal conversations either written contemporaneously or audio-recorded. Entire clinics were attended regardless of presenting complaint, to gain understanding in the context of other long-term conditions. Between consultations practitioners recounted recent eczema consultations. Available documentation was reviewed. Single, semi-structured interviews using a topic guide (box 1)

### Table 1   Demographic details of interview participants

| Role | Gender | Years in current role |
|---|---|---|
| Health visitor | Female | 10 |
| GP | Male | 35 |
| GP trainee | Female | 2 |
| Practice nurse | Female | 31 |
| Practice nurse | Female | 32 |
| Pharmacist | Male | 8 |
| GP trainee | Female | 5 |
| Pharmacist | Female | 12 |
| Pharmacy counter staff | Female | 10 |
| Pharmacy counter staff | Female | 17 |
| GP trainee | Female | 7 |
| GP | Female | 6 |
| GP | Female | 5 |
| Health visitor | Female | 2 |
| Health visitor | Female | 2 |
| Health visitor | Female | 3 |

GP, general practitioner.

were conducted with practitioners from each profession (n=16) (table 1) using maximum variation purposive sampling[38] to ensure a mix of job role and level of experience. A predominance of female participants was reflective of the profile of the healthcare team. The complete data set is summarised in box 2.

Interviews were conducted in the workplace and lasted from 22 to 40 min. Data sufficiency was achieved when no new insights were forthcoming.[39] For completeness documents and websites were reviewed including the National Institute for Health and Care Excellence (NICE) clinical guidance for eczema,[17] the local emollient formulary and the Clinical Knowledge Summary[40] and GP Notebook pages[41] for eczema.

Data collection and analysis were iterative with initial findings being used to guide further collection.[42] Audio-data were professionally transcribed and transcripts read against the recording by the researcher to confirm accuracy. Data analysis was completed independently by the researcher, through the lenses of mindlines and self-management. Transcripts and field notes were read in full to get a sense of the data as a whole, and then manually coded, categorised and merged into themes and annotated with researcher inductive interpretations (see table 2 for worked example). Post theme development, relevant sections of the data were revisited to ensure authentic interpretation and use of participant language.

### Reflexivity

Reflexivity was maintained throughout the study with particular attention being paid to subjectivity and positioning as a nurse and skin health researcher; pre-understandings were consciously set aside.[43]

### Patient and public involvement

Lay people, from an eczema support group, were involved in the development of the research question and in planning the design of the study. They contributed through one meeting and a series of email exchanges.

**Table 2** Example of data analysis process

| Codes (from interview and observational data) | Categories | Theme |
|---|---|---|
| **GP interview**<br>► Eczema 'simple to treat' nothing much has changed over the years – it's bread and butter to us.<br>**HV interview**<br>► Basics are the same, but there's lots of personal preference.<br>**GP interview**<br>► Common complaint 'know by heart'.<br>**Observational data**<br>► Perception from GPs that it's a straightforward condition, treatment is fairly standard and that there is limited need for further knowledge. Intranet rarely used but fairly standard set of resources for GPs. | It's simple to treat | Beliefs about eczema |
| **GP interview**<br>► Software will fire up a message if another product should be used.<br>**Pharmacist interview**<br>► Script Switch – computer tells you if you are prescribing the wrong thing and suggests an alternative.<br>**Observational data**<br>► Belief that guidance is more about cost that research.<br>**Observational data**<br>► Eczema is not a condition that is mentioned in 'learning' interactions such as debriefs. | No need to think too much | |

GP, general practitioner; HV, health visitor.

## RESULTS

Data analysis resulted in three themes: beliefs about eczema, eczema knowledge and approaches to self-management. Each is discussed with examples from the data below.

### Theme 1: beliefs about eczema

Eczema was consistently viewed as a '*bread and butter*' (GP) condition that accounted for many consultations. However, although 19.5% of the practice population was recorded as having some type of eczema few consultations primarily for this condition were observed. Analysis of patient reported reason for GP consultation for a typical week during observation revealed that 26/627 (4.1%) of reasons were skin related with none citing eczema as the primary complaint. No observed face-to-face consultations were primarily for eczema; it was reported as a secondary concern in a small number of GP consultations and more often to HVs in baby clinics. This resulted in eczema necessarily being given limited attention '*it's often a secondary problem and there's only time to deal with one problem per consultation*' (GP). Telephone consultations with GPs were witnessed and patients were observed to consult with pharmacy staff about their eczema. Practitioners mainly viewed eczema as a nuisance condition requiring limited knowledge to treat effectively, '*eczema is simple to treat, nothing much has changed over the years*' (GP) and '*the recipe doesn't change*' (GP).

Some GPs described eczema as a '*catch up*' (GP) consultation when clinics were over-running. GPs and nurses noted the absence of specific external incentives for long-term eczema management and that it was a condition without the '*red flags*' (GP) which trigger treatment escalation or referral. They described treatment options as straightforward involving emollients with or without intermittent topical steroids. Few mentioned calcineurin inhibitors or other available medications. Most practitioners considered emollients to be a homogenous group of preparations all with similar properties, although a few differentiated in terms of viscosity and texture. Pharmacy staff and HVs were familiar with a broader range of emollient products and were more likely to offer suggestions for over-the-counter preparations. This was in part because no HVs in this study were able to prescribe. GPs were reluctant to prescribe topical steroids or other treatments unless absolutely necessary. PNs rarely saw patients with atopic eczema.

Practitioners recognised that eczema could have a negative impact on well-being and quality of life but this was not often reflected in the care offered. Treatment was mainly in reaction to a flare rather than there being a long-term plan of care. Generally patients were able to access regular repeat prescriptions for emollients and practitioners expressed a level of frustration when they presented with a flare having not requested or used the prescribed treatments. Although 'safety netting' was always in place, planned follow-up consultations were not suggested. Empathy for patients was most evident in practitioners who had personal experience of eczema, they articulated a varying level of understanding about the differences between products, regardless of available

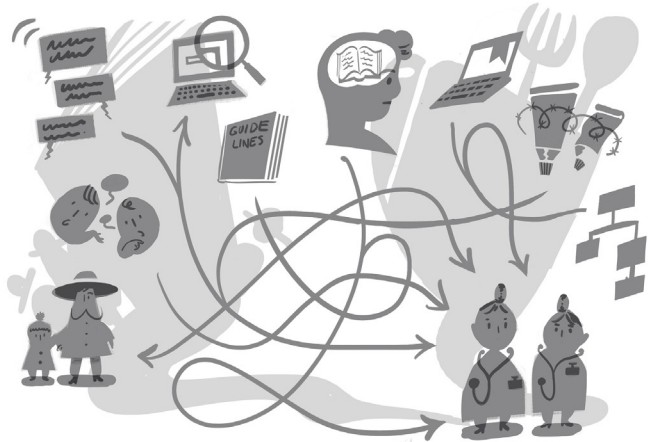

**Figure 1** Practitioner eczema mindlines. Sources of information underpinning practitioner eczema mindlines.

empirical evidence, and the extent to which personal preference influenced concordance. Pharmacy counter staff were the most conversant with the differences between emollient products having tried samples, and they were most likely to share this knowledge with patients/customers verbally and in leaflets.

Although eczema was viewed as a frequent reason for consultation, it was mainly presented as a secondary concern and so dealt with swiftly. Eczema was considered simple to treat with little change over time although practitioners with personal experience of eczema were more aware of the challenges of self-management and tolerant of personal treatment preferences.

### Theme 2: atopic eczema knowledge

Beliefs about eczema influenced the formation of mindlines and for most mindlines were set against a backdrop of eczema being a low priority condition and a perception of unchanging treatment options which were constrained by local prescribing guidelines (figure 1). Many practitioners described atopic eczema as a common conditions for which you '*know (treatment) by heart*' (GP)' and likened his response to using a '*satnav …………you stop thinking, the little NHS boxes (on the computer) tell you what to prescribe*' (GP).

Most practitioners reported that their eczema knowledge was based on their initial education and recognised '*pre-reg derm education was very, very basic*' (GP). A few had completed dermatology placements during GP training but reported seeing little eczema. One experienced GP recounted learning from a consultant, her practice was unchanged as she had '*learnt from a consultant many years ago and never heard anything to contradict it*' (GP). PNs and GPs were aware of available dermatology education but did not attend as it was a low priority and costly, '*there is training but you have to pay*' (PN) and they preferred to '*avoid reps and sponsored sessions*' (GP). HVs reported that skin health was never an educational priority. Pharmacist's knowledge was updated through e-bulletins from different sources and covered only changes in, and availability of,

medications. Only PCS received eczema specific education by attending regular seasonal sessions provided by their employer. Although deemed to be useful, particularly as they tried products and were advised on correct application, the educational experience was sometimes suboptimal as one reported how she was '*shamed into remembering*' (PCS) session content.

Local emollient guidelines underpinned many prescribing decisions so practitioners did not need to think as '*software will fire up a message if another product should be used*' (CP). Changes to guidelines were ascribed to cost and '*what was in vogue*' (GP). Practitioners were not concerned about these changes stating for example, '*aqueous cream, they've gone off that idea for some reason*' (PCS) and '*Zero products are the ones that are currently on trend*' (GP trainee). While some prescribers stuck rigidly to prescribing the cheapest product, '*I try to be good and prescribe the cheaper side of things*' (GP trainee), others were more flexible according to their own or the patient's preference. However, deviations from the formulary were rare on the basis that '*local formulary is very constraining and you'd have to be able to justify why you'd prescribed anything else*' (GP). Exceptions were observed in the baby clinic and in pharmacy practice where patients were often informed about a wider range of emollients that could be purchased over-the-counter. For those who paid a prescription charge this could often be more cost effective. PCS suggested that they were able to advise patients readily as they had '*tried samples so you can tell the customers what they feel like*' (PCS).

Other knowledge sources contributed to eczema mindlines. All staff, with the exception of experienced GPs, used internet searches most commonly the online resources GP Notebook and Clinical Knowledge Summaries. Useful websites were often bookmarked and visited in preparation for a consultation rather than alongside the patient. If information was not located almost immediately the practitioner switched to another website '*we're hard wired for speed now*' (GP) and '*dipped into what's relevant*' (GP) as and when required. None mentioned existing NICE eczema guidance. Local emollient guidelines existed and influenced the prescribing practice of most practitioners, however others were unaware of these and some found them hard to access. GPs and HVs used different emollient guidelines and this caused confusion for patients when they consulted both. A member of pharmacy staff noted the need for '*a synchronised approach so patients don't get confused*' (PCS). Practitioners also experienced confusion when offering advice on treatment application, for example '*treatment is a bit arbitrary – for example should you advise steroid or emollient first?*' (HV).

Practitioners learnt from each other to a limited extent, most often within their professional groups. They recognised '*we learn both good and bad habits from each other*' (PN). Opportunities for shared learning had reduced as there was little time to meet up and in-house teaching for GPs and PNs had '*fallen by the wayside*' (GP) due to staff sickness and pressure of work. One GP reported

'*phoning a friend*', now a consultant dermatologist, when she needed advice. GPs reported learning from trainees during debrief sessions but could not recall ever having discussed eczema. Trainees exhausted all available information sources before seeking advice from a GP. HVs and PNs met more frequently and exchanged knowledge more regularly, although eczema was not a condition of interest.

Practitioners expressed varied views on the value of patient knowledge and experience and the extent to which it influenced care. PNs, HVs and pharmacy staff respectively reported that they routinely '*ask patient what they have tried already*' (PN), '*see what's worked for them*' (HV) and '*listen and learn from customers*' (PCS) and used this information as a basis for treatment advice. Others listened to patients with a degree of scepticism but acquiesced to patient preference, '*patients often have fixed ideas (about emollients) and I try to accommodate these*' (GP). A few were less receptive, for example '*I try to use guidelines and the formulary ………… patient experience stuff can be counterproductive*' (GP trainee) and others suggested that their wider experience overrode the patients personal preferences and experiences '*experience wise I've found a lot of people get on with it (particular emollient)*' (GP) and therefore that was what would be prescribed.

Only the most experienced practitioners spontaneously articulated the existence of tacit knowledge stating, '*it's a perpetual exercise … adding on knowledge and skills*' (GP) and '*built up knowledge over time*' (PN). Others pointed to more concrete sources of knowledge. All practitioners understood reliability of evidence to a greater or lesser extent.

Eczema knowledge was constructed from different sources by individual professions. Nursing and medical staff perceived a limited need to update their knowledge as eczema care was viewed as having changed little over time. Exceptions to this were practitioners who had personal experience of eczema and pharmacy staff who regularly updated their mindlines using informal and formal sources of knowledge.

### Theme 3: approaches to self-management

In principle, all practitioners supported self-management of eczema but recognised the difficulties of achieving this in practice particularly without formal recognition as a long-term condition (LTC). Some practitioners routinely used techniques to support self-management for patients with other LTCs. Strategies included for example, '*finding out patients' expectations*' (PN), '*tailoring knowledge to the person*' (GP), '*start with what the patient understands and then fill in the gaps*' (GP), '*give patients a map of management*' (GP), '*instil confidence*' (GP) and '*reinforce that self-management is good*' (GP trainee). A few GPs used specific techniques such as '*short bursts of CBT*' (GP), '*motivational interviewing techniques ……. compressed to fit in consultation*' (GP) and '*behaviour modification ……….. not a one consultation job*' (GP). Even practitioners who did not articulate using strategies to support self-management integrated

them in practice for many LTCs. However they were rarely observed or discussed in relation to eczema.

Most eczema care was reactive when patients presented with a flare and talk of eczema care was almost exclusively about treatment options. Virtually no attention paid to ensuring that the patient understood the condition and actions they could take to avoid the relentless cycle of flares. The most tangible contribution to self-management was the availability of repeat prescriptions for emollients but advice to use these consistently was lacking. Barriers to self-management were observed, for example the appointment system often precluded patients seeing the same GP over time, so treatment could be altered without the benefit of fully understanding the patient journey to date. Contradictory advice given by practitioners and a lack of faith in patient's ability to judge when they needed to use topical steroids and to use them safely presented significant barriers to successful self-management. Practitioners suggested they '*need to see patients before prescribing (topical) steroids*' (GP). One GP stated that '*sensible*' patients may get steroids on repeat but struggled to define sensible in this context. Pharmacy staff did not recognise their contribution to self-management per se, but recognised the positive impact they had on eczema management through '*actually taking notice of what they're telling me*' (PCS) and perceived '*they do trust me ……………… I'm well known in the local community*' (PCS) and were therefore easy for customers to speak with.

While recognising the need for self-management the fact that eczema is not categorised as a long-term condition limited how much patients were supported to self-manage and at times healthcare systems could hinder attempts.

### DISCUSSION

This study offers new insights into how primary care practitioners construct atopic eczema specific mindlines. Practitioner mindlines are predominantly set against a back drop of eczema being a low priority, due to a combination of not being viewed as an LTC and so lacking external incentives, and the perception of available treatments being standard use of emollients and topical steroids, which changes little over time and is constrained by prescribing guidelines. This led to an assumption that there was little need to amend mindlines. Eczema mindlines were developed early in their career by many practitioners and were relatively static among GPs, PNs and HVs, except for those with direct personal experience of eczema. Mindlines of pharmacy staff were regularly modified through a combination of education provided by their employer, electronic updates from professional bodies and interactions with customers. The latter was particularly influential for the PCSs as they generally had more time to listen and had built up trusting relationships with the customers over time.

This study is one of few to apply mindline theory to a specific condition across a broad range of practitioners. In particular it identifies important differences in the way in which eczema mindlines are developed and so may best be amended for individual practitioner groups. This study conforms with conventions of robust qualitative work in that it is rigorous (coherent and sufficiently well reported to be open to external audit), relevant (enriches understanding of the subject), resonant (resonates with readers experiences and understandings) and reflexive (subjectivity of the author is acknowledged).[44] Limitations include the ethnographer being a lone worker and data analysis being completed by the researcher alone, however this is mitigated by conversations with participants to check understandings. As data were collected in one general practice, findings may not be transferable but the diversity of participants should minimise this risk.[45] Additionally no nurse practitioners were included as, at the time of data collection, none were employed in the practice.

As with the original conceptualisation of Gabbay and le May,[30] practitioner eczema mindlines are composed over time, from a range of evidence sources which rarely embrace direct use of research. Gabbay and le May[28] point to the critical nature of knowledge-in-practice-in-context in which in each context new knowledge is converted by the complex social processes of the socialisation, externalisation, combination, internalisation spiral.[32] Context was central in the formation of eczema mindlines but was informed more by long-held beliefs and national policy than by local context. Key differences in this study are that mindline development has evolved alongside the changing nature of primary care where practitioners, particularly GPs, appear to work more in isolation than as part of a community with 'coffee room chat'[46] appearing much reduced. In parallel, available online resources have spiralled thus potentially reducing the need to confer with others. This challenges the notion that mindlines are heavily reliant on professional interactions.[28] The static nature of eczema mindlines and the beliefs underpinning eczema care meant that they were accessed using fast, automatic, System 1 thinking rather than the more deliberative, conscious, slow and effortful System 2 approach.[47]

Few studies have investigated condition specific mindlines with the exception of a Tanzanian study of malaria diagnosis,[48] however the depiction here is more akin to rules of thumb or heuristics. A comprehensive commentary on mindlines identifies 76 papers categorised as 'in practice', that is studies of how mindlines are developed, many of these used the term to mean consulting with colleagues.[33] A smaller number were faithful to the original Gabbay and le May's conceptualisation but add little by way of new understanding. More recently, Wieringa and colleagues[49] investigated mindlines development in online clinical communities concluding that they offered collective, dynamic settings and suggest implicitly that they may be areas for mindline amendment. While online communities may appeal to some practitioners, this will not be so for all.

In this study eczema was considered low priority. These beliefs are long-standing with surveys suggesting that both patients and practitioners perceive dermatology as a poor relation in healthcare[50–52] and Magin and colleagues[4] describing 'dismissive' and 'unsympathetic' attitudes among GPs. Eczema appears to be considered as 'health problem which is not an illness'[53] and therefore less legitimate and worthy than other conditions. Ambivalence about eczema specific learning was in contrast to a survey which indicated a desire for new knowledge, particularly in the form of education delivered by consultants[54]; inevitably GPs completing the survey would be those with an interest in dermatology. The dermatology community has used many strategies to make research findings accessible to all with limited success.[55] In contrast with this study in which treatment for eczema was viewed as simple others report GPs uncertainty about managing eczema.[56]

Achieving change in primary care practice is challenging, interventions most likely to influence practice demonstrate evidence of benefit, are simple to use and adaptable to local context.[57] The context of eczema mindlines, that it is a low priority condition with a limited repertoire of treatment options, is unlikely to change in the foreseeable future. If, like other LTCs, eczema was recognised in the Quality and Outcomes Framework[58] patients may benefit from the accelerated trends towards systematic management.[59] Practitioners in primary care are expert generalists[60] and are expected to have knowledge of many conditions for which there is wealth of available evidence. This may lead to information overload for which coping strategies are needed. Bate and colleagues[61] describe 'satisficing' that is, curtailing the amount of information gathered to enable them to make a 'good enough' decision.

In many ways it can be argued that treatment of eczema in primary care is relatively straightforward and that amendment of mindlines to adjust thinking about emollients and removal of outdated information about topical steroid use could make a significant change in practice that would improve both patient experience and self-management practices. Brevity and accessibility of information is key as practitioners have been found to judge the usefulness of new knowledge as function of its relevance x validity ÷ by the work needed to access it.[62] It is possible that straightforward messages could be conveyed through media such as aphorisms, 'succinct sayings that offer advice'[63] or actionable nuggets 'knowledge translation tools designed to provide …… concise practical information about the most prevalent and pressing primary care needs of patients'.[64] This approach offers the opportunity to compensate for the loss of professional wisdom through personal communication by transmitting concentrated wisdom and guidance in a different way.[63]

Efforts to amend GPs, PNs and HVs mindlines need to be accessible via rapid System 1 thinking. Interventions should be specific, practical, tailored, relevant and

rapidly delivered information which can readily be assimilated, or as participants in this study described it, a 'no faff' approach. Given their time constraints and information gathering habits, any new information would best be delivered individually rather than in a group setting and available online and possibly in other formats.

The role of the community pharmacist in eczema care is evolving partly in response to Pharmaceutical Services Negotiating Committee guidance on Medicines Use Reviews,[65] New Medicine Service[66] and Minor Ailment Service.[67] Forthcoming changes in availability of emollients on prescription may increase their role further. Pharmacy staff described eczema mindline development as a more collective experience than other practitioners and valued learning from each other and from customers. They may be open to group approaches to update and remove redundant information from their mindlines and this would need to be brokered through both professional and employing organisations.

## CONCLUSION

This ethnographic study provides new understandings about the development of atopic eczema specific mindlines in different practitioner groups in primary care. The outstanding challenge is to find novel, context-specific, simple, pragmatic strategies to revise or modify these mindlines by adding reliable and useful knowledge and by erasing outdated or inaccurate information using strategies that are most appropriate to each profession. Mindline amendment has the potential to improve self-management and quality of eczema care through the delivery of consistent, evidence-based care.

**Twitter** @FCowdell

**Acknowledgements** Thanks to Amanda Roberts who have given invaluable lay feedback on the planning and design of this study. Thanks to James Mycock, Birmingham City University for the mindline illustration and to Professors Hywel William and Stephen Timmons, University of Nottingham for their valuable feedback on earlier iterations of this manuscript.

**Contributors** FC is the sole contributor to this paper.

**Funding** Fiona Cowdell is funded by a National Institute for Health Research, Knowledge Mobilisation Research Fellowship, KMRF-2015-04-004.

**Disclaimer** The views expressed are those of the author and not necessarily those of the NHS, the NIHR or the Department of Health and Social Care.

**Competing interests** None.

**Patient consent for publication** Not required.

**Ethics approval** The study was approved by a National Health Service REC (16/YH/0252). Process consent was used for observation, on each occasion informal conversations were used to re-check participant's willingness to be observed. Patients were informed about the study by practitioners and when necessary the researcher exited individual consultations, either at the request of the patient, the practitioner or using personal judgement, although this was infrequently needed. Written consent was taken for audio-recorded interviews. Interview participants consented to publication of anonymised information.

**Provenance and peer review** Not commissioned; externally peer reviewed.

**Data sharing statement** The data sets generated and/or analysed during the current study are not publicly available as they are not designed to be re-analysed by others but are available from the corresponding author on reasonable request.

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
