## [Reviewer comments · BMJ Open]

ARTICLE DETAILS

TITLE (PROVISIONAL)	Knowledge mobilisation: An ethnographic study of the influence of practitioner mindlines on atopic eczema self-management in primary care in the United Kingdom
AUTHORS	Cowdell, Fiona

VERSION 1 - REVIEW

REVIEWER	Corinna Rea Boston Children's Hospital United States
REVIEW RETURNED	06-Nov-2018

GENERAL COMMENTS	This is an interesting approach to this topic. I am not very familiar with the ethnographic approach, but can see the appeal for this subject. Still, I think a little more rigor is required. More detail is needed about the number of providers in the practice, how the 250 hours were divided, and how much of the results came from the interviews vs the observation. Gabbay and le May include a data summary table in their article, and maybe something similar could be used here. I also found it quite problematic that only one person coded the data. Furthermore, little detail is provided about this analysis. A few additional small comments: -table 2 should at least include gender as this is mentioned in the text. Age could also be helpful.-The author states that lay people were involved in developing the research question, but it is not clear who these lay people are or what their role was-the discussion is very long and should be shortened significantly--the author does not need to include as much detail about the concept of mindlines-the limitations should be emphasized more--only one coder and only one practice. These are significant.
--

REVIEWER	Regina Fölster-Holst Dermatology, University Clinics of schleswig-holstein, Germany
REVIEW RETURNED	18-Nov-2018

GENERAL COMMENTS	Patients with eczema are often treated and advised by non-dermatologists such as general practitioners and pharmacists. Based on observation and interview data the primary care
--

	practitioners see the eczema of low priority and easily to be managed. Management is made of knowledge over time and this is relatively static. That's my experience too. But how could we change that? Some suggestions should be discussed in the manuscript.
--	---

REVIEWER	Parker Magin University of Newcastle, Australia
REVIEW RETURNED	27-Feb-2019

GENERAL COMMENTS	This manuscript reports very interesting findings concerning practitioners' approaches to (I think) atopic eczema. The use of 'mindlines' as a lens for the exploration of this topic has provided a perspective that may prove useful in improving primary care management of atopic eczema. The manuscript presents a good case and rationale for the importance of this as a topic for enquiry. The first issue for the author to address, though, is just what is the condition being explored. This is not specified. The term 'eczema' is used. It seems apparent from the context in a number of places that this is atopic eczema (and the cited references suggest this is so). But it's not stated in the manuscript and in the text (end of second paragraph, 'Theme1: beliefs about eczema' section of Results) it is stated that PNs rarely saw patients with eczema other than older people with varicose eczema. This is at odds with the argument of the rest of the manuscript, including the claim to novelty for the study in that it applies mindline theory to 'a specific condition'. So, it needs to be stated what condition/conditions are being studied and, if it is atopic eczema, the statement regarding varicose eczema isn't relevant to the argument. It isn't clear in the Methods how the pharmacy/pharmacies and baby clinics at which participants worked were selected or what their relationships to the single general practice were. This should be provided along with information of how the participants were recruited. The study found that eczema was considered a 'low priority' condition. As such, some more could be presented in the Introduction and Discussion regarding the evidence for the marked psychological/psychiatric co-morbidity and quality of life impairments associated with atopic eczema (for patients and for their families).
---

VERSION 1 – AUTHOR RESPONSE

Reviewer(s)' Comments to Author:

Reviewer: 1 Reviewer Name: Corinna Rea Institution and Country: Boston Children's Hospital, United States Please state any competing interests or state 'None declared': None declared	
This is an interesting approach to this topic. I am not very familiar with the ethnographic approach, but can see the appeal for this subject. Still, I think a little more rigor is required. More detail is needed about the	Thank you for these comments. The number of providers in the practice varied from day-to-day as, for example, health visitors visited for specific clinics only and some days there were several trainees and on others none. For these

number of providers in the practice, how the 250 hours were divided, and how much of the results came from the interviews vs the observation. Gabbay and le May include a data summary table in their article, and maybe something similar could be used here. I also found it quite problematic that only one person coded the data. Furthermore, little detail is provided about this analysis. A few additional small comments:	reasons adding the number of providers would not be useful. The 250 hours is difficult to divide in a meaningful way and this would not be usual practice in ethnographic studies. However I have added a complete dataset summary for clarification. It's an interesting point about the data analysis. In ethnography the researcher is generally a lone worker, I'm not convinced that having anyone else trying to analyse my field notes would add value. Interview and observational data were treated as a whole entity and therefore attempts to state how much data came from each would not be appropriate. I've set out the method of data analysis in the manuscript "Audio-data were professionally transcribed and transcripts read against the recording by the researcher to confirm accuracy. Data analysis was completed independently by the researcher, though the lenses of mindlines and self-management. Transcripts and field notes were read in full to get a sense of the data as a whole, and then manually coded, categorised and merged into themes. Post theme development, relevant sections of the data were revisited to ensure authentic interpretation and use of participant language". This conforms to the requirements of reporting qualitative data analysis. I've added an illustration of manual data analysis for clarity.
-table 2 should at least include gender as this is mentioned in the text. Age could also be helpful.	Good point, I have added gender to the table. I did not record age of participants, as years in current role was a more useful detail.
-The author states that lay people were involved in developing the research question, but it is not clear who these lay people are or what their role was	Good point, thank you I have amended the manuscript accordingly.  Lay people, from an eczema support group, were involved in the development of the research question and in planning the design of the study. They contributed through one meeting and a series of email exchanges.
-the discussion is very long and should be shortened significantly--the author does not need to include as much detail about the concept of mindlines.	I take your point here but think the discussion around mindlines is essential as most readers will not be familiar with the concept.
-the limitations should be emphasized more--only one coder and only one practice. These are significant	Good point, thank you I have amended the manuscript accordingly.  Limitations include issues of reliability as the ethnographer is a lone worker and data

	analysis was completed by the researcher alone, however this is mitigated by conversations with participants to check understandings. The point regarding one practice is already stated in the sentence  As data was collected in one general practice, findings may not be transferable but the diversity of participants should minimise this risk
Reviewer: 2 Reviewer Name: Regina Fölster-Holst Institution and Country: Dermatology, University Clinics of schleswig-holstein, Germany Please state any competing interests or state 'None declared': no conflict of interest	
Patients with eczema are often treated and advised by non-dermatologists such as general practitioners and pharmacists. Based on observation and interview data the primary care practitioners see the eczema of low priority and easily to be managed. Management is made of knowledge over time and this is relatively static. That's my experience too. But how could we change that? Some suggestions should be discussed in the manuscript.	Thank you, you are absolutely right and I've captured this point in the manuscript section below  The outstanding challenge is to find novel, context-specific, simple, pragmatic strategies to revise or modify these mindlines by adding reliable and useful knowledge and by erasing outdated or inaccurate information using strategies that are most appropriate to each profession. Mindline amendment has the potential to improve self-management and quality of eczema care through the delivery of consistent, evidence-based care. I am now working on methods to amend eczema mindlines and hope to publish in BMJ Open in the future
Reviewer: 3. Reviewer Name: Parker Magin Institution and Country: University of Newcastle, Australia Please state any competing interests or state 'None declared': None declared	
This manuscript reports very interesting findings concerning practitioners' approaches to (I think) atopic eczema. The use of 'mindlines' as a lens for the exploration of this topic has provided a perspective that may prove useful in improving primary care management of atopic eczema. The manuscript presents a good case and rationale for the importance of this as a topic for enquiry.	Thank you, I'm glad you found this manuscript interesting.
The first issue for the author to address, though, is just what is the condition being explored. This is not specified. The term 'eczema' is used. It seems apparent from the context in a number of places that this is atopic eczema (and the cited references suggest this is so). But it's not stated in the manuscript and in the text (end of second	Good point about atopic eczema thank you, I've amended throughout.

paragraph, 'Theme1: beliefs about eczema' section of Results) it is stated that PNs rarely saw patients with eczema other than older people with varicose eczema. This is at odds with the argument of the rest of the manuscript, including the claim to novelty for the study in that it applies mindline theory to 'a specific condition'. So, it needs to be stated what condition/conditions are being studied and, if it is atopic eczema, the statement regarding varicose eczema isn't relevant to the argument.	
It isn't clear in the Methods how the pharmacy/pharmacies and baby clinics at which participants worked were selected or what their relationships to the single general practice were. This should be provided along with information of how the participants were recruited.	Thank you, I've amended for clarity  • Observations were also conducted in a community pharmacy adjacent to the practice, which was used by most patients. • Observation began with the reception team to understand the day-to-day working of the practice. Observation of consultations with GPs, GP trainees and locums, nurses, health visitors in baby clinics, held on the practice premises, and pharmacy staff followed
The study found that eczema was considered a 'low priority' condition. As such, some more could be presented in the Introduction and Discussion regarding the evidence for the marked psychological/psychiatric co-morbidity and quality of life impairments associated with atopic eczema (for patients and for their families).	I completely agree with the sentiment of this point. However I'm reluctant to add much more about the impact of AE, as this is already well known. My aim is to communicate about mindlines and how they may best be amended and I'm keen not to dilute this message.

VERSION 2 – REVIEW

REVIEWER	Corinna Rea Boston Children's Hospital, USA
REVIEW RETURNED	02-Apr-2019

GENERAL COMMENTS	I think this is much improved--more clear and succinct. I ma not familiar with this form of research, but I found it interesting and readable. I would suggest more careful proof reading of the text as there are a few typos.
---

REVIEWER	Fölster-Holst Dermatology of the university clinics of Schleswig-Holstein
-----------------	--

REVIEW RETURNED	30-Mar-2019
-------------

GENERAL COMMENTS	The ethnographic study performed in the United Kingdom used an observation of over 250 hours and interview data to analyse the influence of practitioner mindlines on atopic eczema self-management in primary care. The conclusion is that there are many deficiencies in taking care of patients with atopic dermatitis. I think that this manuscript is worth to be published because It reflects the true care of patients with atopic eczema. My only comment is that it would be nice to know how to improve this situation including education of students, simple guidelines and training for practitioners. This should be addressed more explicitly in the discussion. The ethnographic study performed in the United Kingdom used an observation of over 250 hours and interview data to analyse the influence of practitioner mindlines on atopic eczema self-management in primary care. The conclusion is that there are many deficiencies in taking care of patients with atopic dermatitis. I think that this manuscript is worth to be published because It reflects the true care of patients with atopic eczema. My only comment is that it would be nice to know how to improve this situation including education of students, simple guidelines and training for practitioners. This should be addressed more explicitly in the discussion.
---

REVIEWER	Parker Magin University of Newcastle Australia
REVIEW RETURNED	17-Mar-2019

GENERAL COMMENTS	The author has satisfactorily addressed my original review comments. I think this is a useful addition to the literature in this area.
---

VERSION 2 – AUTHOR RESPONSE

Comment	Response
Reviewer: 3 The author has satisfactorily addressed my original review comments. I think this is a useful addition to the literature in this area.	Thank you
Reviewer: 2 The ethnographic study performed in the United Kingdom used an observation of over 250 hours and interview data to analyse the influence of practitioner mindlines on atopic eczema self-management in primary care. The conclusion is that there are many deficiencies in taking care of patients with atopic dermatitis. I think that this	Thank you I agree that there is a need to change practice and have suggested in my abstract and discussion that this should be approached by changing eczema mindlines “The outstanding challenge is to find novel, profession and context-specific, simple, pragmatic strategies to revise or modify

manuscript is worth to be published because It reflects the true care of patients with atopic eczema. My only comment is that it would be nice to know how to improve this situation including education of students, simple guidelines and training for practitioners. This should be addressed more explicitly in the discussion.	practitioner mindlines by adding reliable and useful knowledge and by erasing outdated or inaccurate information thus potentially improve quality of eczema care". This is the focus of another paper that is currently under review with BMJ Open
Reviewer: 1 I think this is much improved--more clear and succinct. I ma not familiar with this form of research, but I found it interesting and readable. I would suggest more careful proof reading of the text as there are a few typos.	Thank you, I have corrected typos and added punctuation.